# SARS-CoV-2 Seroprevalence Survey in People Involved in Different Essential Activities during the General Lock-Down Phase in the Province of Prato (Tuscany, Italy)

**DOI:** 10.3390/vaccines8040778

**Published:** 2020-12-19

**Authors:** Vieri Lastrucci, Chiara Lorini, Marco Del Riccio, Eleonora Gori, Fabrizio Chiesi, Gino Sartor, Beatrice Zanella, Sara Boccalini, Angela Bechini, Francesco Puggelli, Paolo Bonanni, Guglielmo Bonaccorsi

**Affiliations:** 1Department of Health Sciences, University of Florence, 50134 Florence, Italy; chiara.lorini@unifi.it (C.L.); beatrice.zanella@unifi.it (B.Z.); sara.boccalini@unifi.it (S.B.); angela.bechini@unifi.it (A.B.); paolo.bonanni@unifi.it (P.B.); guglielmo.bonaccorsi@unifi.it (G.B.); 2Global Health Center, Meyer Children’s University Hospital, 50139 Florence, Italy; 3Medical Specialization School of Hygiene and Preventive Medicine, University of Florence, 50134 Florence, Italy; marco.delriccio@unifi.it (M.D.R.); eleonora.gori@unifi.it (E.G.); fabrizio.chiesi@unifi.it (F.C.); gino.sartor@unifi.it (G.S.); 4Management Department, Meyer Children’s University Hospital, 50139 Florence, Italy; francesco.puggelli@meyer.it

**Keywords:** SARS-CoV-2, COVID-19, serosurvey, seroprevalence, healthcare workers, essential activities, Italy

## Abstract

Serosurveys may help to assess the transmission dynamics in high-risk groups. The aim of the study was to assess the SARS-CoV-2 antibody seroprevalence in people who had performed essential activities during the lock-down period in the Province of Prato (Italy), and to evaluate the risk of exposure to SARS-CoV-2 according to the type of service. All the workers and volunteers of the Civil Protection, employees of the municipalities, and all the staff of the Health Authority of the Province of Prato were invited to be tested with a rapid serological test. A total of 4656 participants were tested. SARS-CoV-2 antibodies were found in 138 (2.96%) cases. The seroprevalence in health care workers, in participants involved in essential support services and in those who worked from home were 4.1%, 1.4% and 1.0%, respectively. Health care workers experienced higher odds of seropositivity (OR 4.38, 95%CI 2.19–10.41) than participants who were assigned to work-from-home; no significant seropositivity differences were observed between support services and work-from-home groups. A low circulation of SARS-CoV-2 was observed among participants performing different essential activities. Findings highlighted the risk of in-hospital transmission in healthcare workers and that community support services may increase the risk of seropositivity to a limited extent in low incidence areas.

## 1. Introduction

At the end of 2019, several cases of severe pneumonia of unknown origin were identified in Wuhan (Hubei, China); the etiological agent of this unidentified pneumonia has been subsequently confirmed as a novel coronavirus [1]. The novel virus was classified as severe acute respiratory syndrome coronavirus 2 (SARS-CoV-2) and the related disease as COVID-19 [2,3]. The number of cases started to exponentially increase across the Hubei province, and then the outbreak quickly spread in many other countries; in March 2020, the World Health Organization declared the COVID-19 pandemic [4]. Italy was one of the first Western countries severely affected by the coronavirus pandemic. After the confirmation of the first two cases of infection in a couple of Chinese tourists from Wuhan (31 January 2020), the outbreak started in the Northern Italy, with the first autochthonous Italian case confirmed on 21 February 2020 [5]. However, SARS-CoV-2 was already circulating undetected at the end of 2019. Indeed, a study carried out by the Italian Institute of Health documented the presence of the SARS-Cov-2 in untreated wastewaters of different geographic regions of Italy during December 2019 [6]. 

At first, the epidemic mainly affected the northern and central parts of the Country, and then spread nationwide. The Italian Government adopted—based on the advice of an expert committee—a three-phase strategy to contain the circulation of SARS-CoV-2. The first phase (i.e., the “lock-down” phase) officially started on March 11th and involved strict containment measures; all non-essential services and activities (including schools) were suspended, and a “stay-at-home” order was imposed, banning all non-essential travel and contact with people. Whenever possible, companies were encouraged to shift from work-from-office to work-from-home in both private and public sectors [7]. After the adoption of these measures, the epidemic showed a decreasing trend in the number of cases, deaths and hospitalizations; from May 2020, it was possible to enter phase two and subsequently into phase 3. These phases allowed the gradual reopening of services and businesses as well as the lifting of the stay-at-home order [8,9,10]. As of today, more than one million people in Italy have been infected with SARS-CoV-2 and at least 44,139 people have died [11]; the spread of COVID-19 is likely larger than officially reported [12]. Furthermore, 55,229 cases occurred in healthcare workers [13].

Healthcare workers as well as all the workers and volunteers engaged in essential activities during the most severe phase of the epidemic were potentially at higher risk of exposure to infection. Lack of information on the prevalence and risk of infection in these population groups may have hampered the proper planning of public health measures. Although it is unrealistic to test the whole population, seroprevalence studies may help in evaluating the transmission dynamics and in assessing how prevalent the infection may be in selected high-risk groups. In the Italian scenario, the Ministry of Health supported by the Scientific Committee has recognized the importance of carrying out serosurveys in order to assess the spreading of SARS-CoV-2 [14]. The first results from SARS-CoV-2 serosurveys in health care workers have begun to appear in the literature [15,16,17], but very limited research has examined other categories of essential workers or compared the risk of infection with that of the general population [18]. 

Tuscany Region (central Italy) and, in particular, the Province of Prato—the setting of the present study—was characterized by a slower epidemic trend than the Italian average [19]. At the end of April 2020, the Tuscany Region started one of the largest mass screening initiatives through the application of rapid serological tests to assess the spread of SARS-CoV-2 in the population. At the beginning, the healthcare workers were the main population target of the study [20], then the test was offered to other professional categories such as workers and volunteers of essential services opened during the lock-down period [21,22]. The Province of Prato was among the first provinces of Tuscany to start the serological survey. This area of Tuscany has strong connections with China as it hosts one of the biggest European Chinese community [23].

Therefore, the aim of the present study is to assess the SARS-CoV-2 antibody seroprevalence in individuals who have performed essential activities during the lock-down period and to evaluate the risk of exposure to SARS-CoV-2 according to the type of service performed. In particular, the study evaluated the seroprevalence of SARS-CoV-2 antibodies in a representative sample of workers/volunteers of the local health authority (LHA), Civil Protection, and municipalities of the Province of Prato.

## 2. Materials and Methods 

The study was approved by the Ethics Committee of the Area Vasta Toscana Centro (Comitato Etico Regionale per la Sperimentazione Clinica della Regione Toscana, Sezione Area Vasta Centro, 17470_oss) and was conducted according to the principles described in the Declaration of Helsinki.

### 2.1. Setting and Study Population

The study had a cross-sectional design and was carried out in a population-based sample. The study was conducted in the Province of Prato (Tuscany, Italy) in May 2020. The study sample was composed of people aged 18 years or older involved in different activities considered essential during the lock-down period. In particular, the following population groups were invited to participate in the study: workers of the Local Health Authority (LHA), workers and volunteers of the Civil Protection, and public employees of the Province of Prato and of the municipalities of Cantagallo, Carmignano, Montemurlo, Poggio a Caiano, Vaiano, and Vernio. All the people belonging to these population groups in the Province of Prato were invited to participate.

### 2.2. Data Collection and Measurements

All the eligible persons were invited to voluntarily join the study by the head of each service, who provided them with an informative letter reporting the aim of the survey. On the testing day, the participants were invited to fulfill the informed consent form, and basic demographic data were collected. 

The SARS-CoV-2 antibody prevalence was assessed through a qualitative analysis using the rapid serological test LYHER Novel Coronavirus (2019-nCoV) IgM/IgG Antibody Combo Test Kit (Colloidal Gold) (Hangzhou Laihe Biotech Co., Ltd, Hangzhou, China), according to the manufacturer’s specifications. The tests were performed at the point-of-care by a team composed by physicians and nurses; antibodies were tested in whole blood. In particular, testing occurred immediately after specimen collection: two drops of whole blood (20 μL) followed by two drops (about 100 μL) of specimen diluent were added to the test well. Test results were read and recorded after ten minutes by the physicians. The test was considered as positive whether or not the IgM and/or IgG band were positive. When no control line appeared or in case of difficulties in interpreting the results, the test was immediately repeated.

### 2.3. Statistical Analysis

Participants were classified in three groups according to the type of essential service carried out during the lock-down period. In particular, all the healthcare workers of the LHA (i.e., physicians, nurses, midwifes, physiotherapists, and patient-facing technicians) were included in the “health service” group; the support staff of the LHA (i.e., personnel employed in essential maintenance, information technologies, administrative, and facilities services, and no-patient-facing technicians), workers/volunteers of the Civil Protection (in Italy the Civil Protection fulfills logistic, organizational, telecommunication, and technical emergency support tasks) and the Provincial/Municipal staff employed in essential community services (i.e., road and building maintenance, environment safeguarding, and essential administrative services) were included in the “support service” group. This group was further explored by separately considering the LHA support services and community support services (i.e., workers/volunteers of the Civil Protection and of the Municipalities/Province). Lastly, LHA and Municipalities/Province workers who were assigned to work-from-home (i.e., administrative profiles) were included in the “work-from-home” group.

Data were described as percentage or as median and inter-quartile range. Estimates of antibody prevalence were reported as mean and 95% CI. Positive participants were defined as those with a positive response either for IgM or for IgG, or for both.

Normality was assessed by using the Shapiro Wilk test. Univariate analyses were performed including an evaluation of association using the Chi2 test or Fisher’s exact test (α = 0.05 and 95% CI) for categorical variables while the Mann–Whitney U and Kruskal–Wallis tests were used for non-normally distributed continuous data.

A multiple logistic regression model was fit using the result of the rapid serological test as dependent variable, and the other variables (i.e., sex, age, and the categories of service performed) as independent variables. 

The statistical analyses were conducted using RStudio 1.2.5033 (RStudio Team, 2019. RStudio: Integrated Development for R. RStudio, Inc., Boston, MA, USA; URL: http://www.rstudio.com). An alpha value of 0.05 was considered significant.

## 3. Results

The present study included 4656 participants, with a study participation rate of 95.5%; the median age of the sample was 49.0 years (IQR 38.0–57.0). Males and females numbered, respectively, 1532 and 3124 and represented 32.9% and 67.1% of the study population, respectively. 

Most of the included participants (3091; 76.7%) were from the LHA, 1039 were volunteers of the Civil Protection (11.6%), and 526 (6.1%) were employees of the municipalities and of the Province of Prato.

Sociodemographic characteristics and SARS-CoV-2 seroprevalence for each group are shown in Table 1. The median age of the sample was 49 years (IQR 38–57), and males represented 32.9% of the sample. A total of 138 (2.96%) participants tested positive for SARS-CoV-2 antibodies. No age or sex differences were found in the seroprevalence of antibodies against SARS-CoV-2 at univariate analysis.

According to the type of service performed during the lock-down phase, 2828 (60.7%) participants were classified in the “health service” group, 1103 (23.7%) participants were in the “support service” group, and 725 (15.6%) participants were in the “work-from-home” group. The Kruskal–Wallis and Chi2 tests showed that there were significant age and sex differences between the different groups (*p* < 0.01). As far as the seroprevalence in the different service groups is concerned, statistically significant differences (*p* < 0.01) were found between the type of service and the seropositivity, with participants of the “health service” group registering the highest seroprevalence (4.1%) and those of the “LHA support service” group registering the second highest seroprevalence (1.7%).

A multiple logistic regression model was fitted considering the test result (positive vs. negative) as a dependent variable. Odds ratios (OR) are shown in Table 2. No significant differences in the likelihood of being SARS-CoV-2 seropositive were found for age and sex. As for the relationship between SARS-CoV-2 seropositivity status and type of service performed during the lock-down, participants belonging to the health service group had an increased odds ratio of being seropositive compared with participants in the work-from-home group (OR 4.38; 95%CI 2.19–10.41; *p* < 0.01). No significant differences were found in likelihood of being SARS-CoV-2 seropositive between participants belonging to the support services groups (community and LHA support services groups) and those in the work-from-home group.

## 4. Discussion

From 11 March 2020 to 1 May 2020, Italy experienced a total lock-down of the whole country as an emergency measure to counteract the rapid increase of COVID-19 cases within the national boundaries. To better comprehend the effective circulation of SARS-CoV-2 in the Italian population and in particular in those groups of people potentially more exposed to COVID-19 cases, serological rapid tests were recommended to be used among high-risk groups. The aim of the current study was to assess and compare the seroprevalence of antibodies against SARS-CoV-2 among different groups of participants involved in essential activities during the lock-down period in the Province of Prato (Tuscany, Italy). Study participants were tested in the period immediately after the lifting of the strict containment measures (i.e., May 2020). Results of our study showed that the overall seroprevalence was 2.96% among the 4656 included participants. As far as the seroprevalence according to the type of essential service performed is concerned, the healthcare workers had a significantly higher seroprevalence than the other two groups (participants who were assigned to the work-from-home and those in the support service groups). No significant differences were observed in the seroprevalence between participants in the support service groups and those assigned to work from home. 

No age or sex differences were found in the seroprevalence of antibodies against SARS-CoV-2. These results are in line with what emerged in previous researches [17,24,25,26,27,28]. 

Results showed that the seroprevalence of the “health service” group was 4.1%; this percentage is in line with the ones found in health care workers of other Italian regions [16] and European Countries [17,18,29]. However, results from other studies reported a large range in the levels of seroprevalence in health care workers, with some studies reporting seroprevalence differences even within small geographical areas [15,17,30,31]. These differences in the seroprevalence in healthcare workers may reflect the specific COVID-19 epidemic situations of the geographical areas in which the studies were performed; furthermore, they may be the result of different access to personal protective equipment and proper training. Lastly, differences in the hospital buildings in terms of space and facilities dedicated to infection control have to be considered while evaluating the seroprevalence in healthcare workers from different areas. 

Although the seroprevalence registered in health care workers was relatively low, it was significantly higher than that observed in participants working from home (1%). The seroprevalence that resulted in the work-from-home group may be considered—to a certain extent—a proxy for the seroprevalence level in the general population of the study area, since participants of this group were confined to their home as all the other population groups not involved in essential activities. This difference in the seroprevalence between healthcare workers and the background population level suggests a relatively low circulation of SARS-CoV-2 in the study area during the first pandemic wave (spring 2020). This result was also supported by the low cumulative incidence of COVID-19 cases registered in the municipalities included in the study [26,32]. Moreover, it highlights that frequent exposure to patients in the context of a COVID-19 referral hospital was probably the main risk factor for seroconversion in healthcare workers when this study was conducted. The above assumptions are also confirmed by the relatively low seroprevalence registered in the community support service group (1%). Indeed, although participants in this group were involved in various social and support activities with other people during the lock-down period, they resulted in having a similar seroprevalence to the work-from-home group, presumably because they were assisting communities of low incidence areas. 

Only a few studies have reported a comparison between the seroprevalence of healthcare workers and the background population of the area [17,18]. Similar to our findings, results of these studies showed the presence of high seroprevalence among healthcare workers and low seroprevalence in the background population of the hospitals’ community area, thus suggesting in-hospital transmission. In this regard, it should be highlighted that our study has measured the seroprevalence at the tail end of the first wave of the pandemic in Italy, and therefore the higher seroprevalence in healthcare workers also reflects the less systemic and rigorous use of personal protective equipment and the unpreparedness of the healthcare organizations during the first phase of the epidemic. Furthermore, at that time, the prevalence of asymptomatic infections was generally underestimated and the definition of suspected cases to be tested included only symptomatic individuals. Today, the risk of in-hospital transmission of SARS-CoV-2 among health care workers may be less significant, since hospitals have adopted all the precautions to counteract and contain the SARS-CoV-2 transmission.

Within the support service group, it was possible to observe that participants employed in essential community support services registered a seroprevalence level identical to the one of participants working from home, while participants working in the LHA support services showed a substantially higher—but not significant—seroprevalence. Although there was a lack of statistical significance, it is important to highlight this tendency as the relatively limited sample size of this subgroup may have limited the statistical power of the analyses. Since other studies have also highlighted a relatively high seroprevalence in workers of the support services of health care organizations [33], larger studies are needed to further explore the risk of SARS-CoV-2 exposure in this service category.

The present study has several strengths and limitations. This is one of the first studies assessing the risk of SARS-CoV-2 seropositivity according to the type of essential service performed during the lock-down period in Italy. Furthermore, the study sample well represents the selected population groups of the considered geographical area, since almost all the eligible persons agreed to participate in the study. Therefore, selection bias can be reasonably excluded, and the results can be considered representative of the entire study area for the selected population groups. As far as the study limitations are concerned, an accuracy analysis of the rapid serological test applied in this study was not performed as additional molecular data in case of a negative serological result were not collected. However, the test used in the study was based on an immune colloidal gold technique to detect IgM/IgG antibodies against the novel coronavirus (in conformity with Directive 98/79/EC) and, as declared by the manufacturer, had a good specificity and sensitivity (respectively, 99.23% and 98.31% for IgG, and 99.23% and 95.73% for IgM) and no cross-reactivity with proteins of other common viruses, legionella, antinuclear antibody (ANA), and rheumatoid factor. The test performance was further confirmed by an FDA independent clinical agreement validation (IgM/IgG-specificity: 98.8%, and IgM+ or IgG+ total sensitivity: 100%; no cross-reactivity described) [34]. Nevertheless, considering the specificity of the test reported by the FDA independent analysis, it should be pointed out that the seroprevalence may be overestimated especially in the low incidence groups, in which the percentage of positive readings could be more affected by false positive issues. Besides, it should be underlined that participants classified in the support service group were either working or volunteering in an essential activity that do not provide a direct exposure to patient contact. However, it was not possible to further detail the activities performed during the lock-down period in this group due to the scattered nature of tasks and services reported. Lastly, as the study did not examine the exposure to cases, preventive behaviors, and tasks in which participants were involved, it was not possible to characterize risk factors associated to seropositivity in any population groups.

## 5. Conclusions

In conclusion, a relatively low prevalence of antibodies against SARS-CoV-2 was observed among participants involved in different essential activities during the lock-down period in the early stage of the COVID-19 epidemic in the Province of Prato (Tuscany, Italy). Participants involved in essential community support services showed a similar seroprevalence to those who worked from home, while the seroprevalence registered among healthcare workers was significantly higher compared with those who worked from home. These results highlighted a relatively low circulation of SARS-CoV-2 in high exposure risk population groups. Furthermore, our findings suggest that in-hospital transmission was the main risk factor for seropositivity in healthcare workers and that community support services during the epidemic may increase the risk of seropositivity to a limited extent in a low incidence area. Rapid antibody screening for SARS-CoV-2 of large population groups is essential to control the epidemic and can help to monitor the transmission dynamics and evaluate the effectiveness of infection control measures.

## Figures and Tables

**Table 1 vaccines-08-00778-t001:** Demographic characteristics and SARS-CoV-2 seroprevalence of the study population.

Type of Service	N (% of the whole sample)	N of Males (% within the Group) *	Median Age, Years (IQR) #	N of Positive Participants (% of within the Group) *	% Positive Participants 95% CI
Work-from-home	725 (15.6%)	248 (34.2%)	49.0 (39.7–56.0)	7 (1.0%)	0.3%–2.0%
Support Service (total)	1103 (23.7%)	481 (43.6%)	50.0 (36.0–61.0)	15 (1.4%)	0.7%–2.2%
*Community support service*	*502 (10.8%)*	*327 (65.1%)*	*53.3 (34.3–64.5)*	*5 (1.0%)*	*0.3%–2.3%*
*Local health authority support service*	*601 (12.9%)*	*154 (25.6%)*	*48.0 (37–59.0)*	*10 (1.7%)*	*0.8–3.0%*
Health Service	2828 (60.7%)	803 (28.4%)	48.0 (38.0–56.0)	116 (4.1%)	3.4%–4.9%
Total sample	4656 (100%)	1532 (32.9%)	49.0 (38.0–57.0)	138 (3.0%)	2.5%–3.5%

* *p* < 0.01 Chi2 Test; # *p* < 0.01 Kruskal–Wallis test.

**Table 2 vaccines-08-00778-t002:** Multiple logistic regression model with test result (positive vs. negative) as dependent variable.

Variables	OR	95% Confidence Intervals	*p*
Age ^1^	1.00	0.99–1.02	0.84
*Sex*			
Male (Ref.)	1	-	-
Female	1.01	0.70–1.48	0.97
*Type of service*			
Work-from-home (Ref.)	1	-	-
Community support service	1.03	0.30–3.26	0.96
Local health authority support service	1.73	0.66–4.80	0.27
Health service	4.39	2.19–10.43	<0.01

^1^ Continuous variable.

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
