# Peer review of "SARS-CoV-2 Seroprevalence Survey in People Involved in Different Essential Activities during the General Lock-Down Phase in the Province of Prato (Tuscany, Italy)"

_vaccines, 2020, doi:10.3390/vaccines8040778_

Round 1

Reviewer 1 Report

In their manuscript, Lastrucci et al. describe the results of a serological survey conducted in the province of Prato, Italy. This serological screening was performed in order to assess the spread of SARS-CoV-2 in the local population in May 2020, following a strict lock-down period.  The manuscript reports SARS-CoV-2 antibody seroprevalence in three different groups, designated according to the type of service they performed during the lock-down period: “health service” group, “support service” group, and “work-form-home” group. A higher rate of seropositivity was observed in the “health service” group (4.1%), while the two other groups displayer lower levels (1.4% and 1%).

The study included a total of 4,656 individuals recruited among institutions Local Health Authority, Civil Protection, and Municipalities and of the Province of Prato. With a high participation percent of 95.5% among the employees of the three institutions, the study design seems to represent the studied population well, with some potential to draw broader conclusions based on the results. The author’s aim to identify differences in SARS-CoV-2 antibody seroprevalence depending on the duties performed by the three groups is of interest.

The findings reported in this study regarding the highest-seroprevalence group (“health service”) are similar to previous reports from Italy and other European countries. However, since the study relies on the exclusive use of one rapid antibody test, LYHER Novel Coronavirus (2019-nCoV) IgM/IgG Antibody Combo Test Kit (Colloidal Gold) (Hangzhou Laihe Biotech Co., Ltd), the incidence rates reported, particularly for the low-incidence groups, may be inaccurate. Herein lies the main limitation of the paper: in order to present their work as a comparative analysis between the three groups as the authors do, they should confirm positive readouts, especially in the two low-incidence groups, using alternative methods. If the authors can strengthen their analysis, and confirm that their readings are accurate, this work may be suitable for publication in Vaccines.

Major comments:

Major comment 1. Authors should confirm that the 1% and 1.4% seroprevalence readings observed in the two low-incidence groups are not, in large part, due to specificity issues with the rabid antibody test. FDA-reported Independent Clinical Agreement Validation of the test used in this study reports a false positive rate of approximately 1 false positives per 80 known-negative samples, which indicates that majority (or even all) of the positive readings in both lower incidence groups (7 among 725 and 15 among 1,103, respectively) could be false positives. The authors should confirm positive readings with an alternative method, preferably for all three groups.

Major comment 2. Sensitivity and specificity values for the rapid antigen test used should be reported, and taken into account in the analysis.

Major comment 3. More details on how the rabid serological assay was performed should be given. How were patient samples prepared for the assay? Where they tested immediately or stored? How were they stored? How were the results read?

Major comment 4. The tasks of individuals in the three groups could be better defined. In particular, what comprises “support services”? Other studies (eg. doi: 10.1136/thoraxjnl-2020-215414) report significant differences in risk to exposure between heathcare housekeeping tasks vs. certain medical tasks. A deeper look into the tasks may yield further insight and strengthen the paper.

Major comment 5: On lines 165-166, “No significant difference was found in likelihood of being SARS-CoV-2 seropositive between participants belonging to the health services group and those in the work-from-home group.” Do the authors mean ‘support services’ group here rather than ‘health services’?

Minor comments:

Minor comment 1. It would be more appropriate to call the individuals taking part in the study participants rather than subjects.

Minor comment 2. Minor polishing of the language would benefit the paper. Examples: article ‘the’ is occasionally misplaced, sentences on line 39 should be fixed, and on line 22, the authors state their intent is to measure ‘risk of seropositivity’, when they most likely mean risk of exposure to SARS-CoV-2.

Author Response

Major comments:

  • Major comment 1. Authors should confirm that the 1% and 1.4% seroprevalence readings observed in the two low-incidence groups are not, in large part, due to specificity issues with the rabid antibody test. FDA-reported Independent Clinical Agreement Validation of the test used in this study reports a false positive rate of approximately 1 false positives per 80 known-negative samples, which indicates that majority (or even all) of the positive readings in both lower incidence groups (7 among 725 and 15 among 1,103, respectively) could be false positives. The authors should confirm positive readings with an alternative method, preferably for all three groups.

Reply: We agree with the reviewer that the confirmation of the positive reading with an additional method would have led to a better accuracy in the assessment of seropositivity, especially in the low incidence groups. This is an issue that affected several other serosurveys carried out during the first wave of the COVID-19 pandemic, see, for example, the studies of Iversen et al, 2020 or Sood et al, 2020, which employed tests of similar or lower performances. Unfortunately, also our study protocol did not comprise a validation of the test results with the use of other serological methods, so we are not able to provide a confirmation of the positive readings with an alternative method. At the moment in which the study was carried out, blood samples were not taken and stored for further analyses; and performing additional analyses on positive participants months after the conclusion of the study will led to unreliable results. Indeed, it is unknown for how long antibodies persist following the infection; furthermore, in case of a confirmed seropositivity, it would not be possible to ascertain whether SARS-CoV-2 antibodies were developed during the study period or after its conclusion.

We have clearly acknowledged this issue as a study limitation in the discussion section of the revised manuscript and reported the test performance as evaluated by the manufacturer and by the FDA independent validation in the discussion section of the revised manuscript (please see pg 7 ll 267-277 of the revised manuscript with tracked changes).

However, examining the results of the FDA analyses on the performance of the authorized serology tests (available on: https://www.fda.gov/medical-devices/coronavirus-disease-2019-covid-19-emergency-use-authorizations-medical-devices/eua-authorized-serology-test-performance), it should be pointed out that the test used by our study resulted to have one of the highest combined specificity (98.8%) and the second best  ratio of sensibility (100%) and specificity out of the seventeen authorized serology tests assessing both IgM and IgG. Furthermore, it should be noted that the specificity was assessed in 80 antibody-negative sample, while a much larger evaluation of our test carried out by the same FDA reported a 99.43% of negative percent agreement: 347 out of 349 negative specimens (239 of them were from patients with coronavirus infections not caused by SARS-CoV-2, and 110 of them were retrospective frozen specimens collected before August 2019) resulted to be IgG and IgM negative. On top of that, our test showed no-cross reactivity in 137 sample (data available on https://www.fda.gov/media/139410/download).

Lastly, it should be underlined that the potential issue of specificity may have led to an overestimation of the seroprevalence, particularly in the two groups that already showed the lower seroprevalence levels; in the worst case scenario this effect would actually strengthen the findings of the comparative analysis among groups and further confirm the study conclusion of a lower risk in the “work-from-home” and “support service” groups.

  1. Iversen, K.; Bundgaard, H.; Hasselbalch, R.B.; Kristensen, J.H.; Nielsen, P.B.; Pries-Heje, M.; Knudsen, A.D.; Christensen, C.E.; Fogh, K.; Norsk, J.B., et al. Risk of COVID-19 in health-care workers in Denmark: an observational cohort study. The Lancet Infect Dis. 2020, Aug 3, S1473-3099(20)30589-2. doi: 10.1016/S1473-3099(20)30589-2.
  2. Sood N, Simon P, Ebner P, Eichner D, Reynolds J, Bendavid E, Bhattacharya J. Seroprevalence of SARS-CoV-2-Specific Antibodies Among Adults in Los Angeles County, California, on April 10-11, 2020. JAMA. 2020 Jun 16;323(23):2425-2427. doi: 10.1001/jama.2020.8279. PMID: 32421144; PMCID: PMC7235907.
  • Major comment 2. Sensitivity and specificity values for the rapid antigen test used should be reported, and taken into account in the analysis.

Reply: as suggested, we have reported sensitivity and specificity values of the rapid antigenic test as stated by the manufacturer; furthermore, we have also reported sensitivity and specificity values of the test as assessed by the FDA- reported Indipendent Clinical Agreement Validation and discussed their implications in the revised manuscript (please see pg 7 ll 267-277 of the revised manuscript with tracked changes).

  • Major comment 3. More details on how the rabid serological assay was performed should be given. How were patient samples prepared for the assay? Where they tested immediately or stored? How were they stored? How were the results read?

Reply: as suggested, we have provided further details concerning how the serological assay was performed in the Materials and Methods section of the revised manuscript (please see pg 3 ll 117-123 of the revised manuscript with tracked changes).

  • Major comment 4. The tasks of individuals in the three groups could be better defined. In particular, what comprises “support services”? Other studies (eg. doi: 10.1136/thoraxjnl-2020-215414) report significant differences in risk to exposure between heathcare housekeeping tasks vs. certain medical tasks. A deeper look into the tasks may yield further insight and strengthen the paper.

Reply: As suggested, the tasks of individuals in the three groups have been better defined in the Materials and Methods section of the revised manuscript (please see pg 3 ll 127-138 of the revised manuscript with tracked changes). Please note that housekeeping tasks were not comprised in the support services as this service is outsourced to private companies by the LHA. Furthermore, considering the study suggested by the reviewer, we have decided to further analyze the support service group as composed by two distinct sub-groups, namely “community support service” and “support service of the Local Health Authority (LHA)”. The method section was modified accordingly in the revised manuscript (please see pg 3 ll 136-138 of the revised manuscript with tracked changes); results of these analyses were described in the result section (please see pg 4 ll 174-175; pg 5 ll 185-187; and table 1 and 2 of the revised manuscript with tracked changes) and commented in the discussion section of the revised manuscript (please see pg 7 ll 250-258 of the revised manuscript with tracked changes).

  • Major comment 5: On lines 165-166, “No significant difference was found in likelihood of being SARS-CoV-2 seropositive between participants belonging to the health services group and those in the work-from-home group.” Do the authors mean ‘support services’ group here rather than ‘health services’?

Reply: Thanks for noticing this, we have corrected the error in the revised manuscript (please see pg 5 ll 186-187 of the revised manuscript with tracked changes)

Minor comments:

  • Minor comment 1. It would be more appropriate to call the individuals taking part in the study participants rather than subjects.

Reply: as suggested, we have replaced “subjects” with “participants”, “people”, “individuals” or “persons”, as appropriate, throughout the manuscript.

  • Minor comment 2. Minor polishing of the language would benefit the paper. Examples: article ‘the’ is occasionally misplaced, sentences on line 39 should be fixed, and on line 22, the authors state their intent is to measure ‘risk of seropositivity’, when they most likely mean risk of exposure to SARS-CoV-2.

Reply: The manuscript has been carefully revised to improve the grammar and readability; the sentences on line 22, and on line 39 have been fixed in the revised manuscript (please see pg 1 ll 22; and pg 1 ll 39-41 of the revised manuscript with tracked changes).

Reviewer 2 Report

The work is interesting, methodologically adequate. In addition, the authors conducted an investigation into covid-19 in population groups that carried out essential activities during this pandemic. However I have the following comments.

I. Major Comments
1. The analytical methodology is adequate, however the description of the sample is very limited. I suggest including:
Weight, height, BMI, chronic diseases, etc. This information is important for a better understanding of the study.

II. Minor comments:
1. Improve the writing of the study objective.

Author Response

  • Major Comments
    The analytical methodology is adequate, however the description of the sample is very limited. I suggest including:
    Weight, height, BMI, chronic diseases, etc. This information is important for a better understanding of the study.

Reply: Unfortunately, we are not able to provide further descriptive variables for the whole sample. Indeed, the large sample size, the limited and precise study time-frame (immediately after the lock-down period), and the emergency period in which the study was carried out did not allowed to collect further descriptive characteristics from health care workers. The characteristics of the other sample subgroups (i.e. workers and volunteers of the Civil Protection, and the public employees of the municipalities) will be deeply analyzed in a future publication.

  1. Minor comments:
    1. Improve the writing of the study objective.

Reply: as suggested the writing of the study objective has been improved in the abstract (please see pg 1 ll 22 of the revised manuscript with tracked changes) and the introduction section (please see pg 2 ll 86-91 of the revised manuscript with tracked changes).

Round 2

Reviewer 1 Report

The authors have made a good effort in addressing issues relating to assay sensitivity and specificity in the manuscript text. In the absence of experimental validation, it remains possible that a significant quantity of the positive test results in low-incidence groups are due to false positives, but to the author’s credit this caveat is clearly stated in the revised discussion. I agree with the authors that the trend of higher seropositivity observed in healthcare workers vs. other groups appears authentic and comprises an important observation.

In line with their revised analysis, I suggest that the authors should revise the ending of their abstract:

(Abstract: “Findings highlighted the risk of in-hospital transmission in healthcare workers and that community support services may increase the risk of seropositivity to a limited extent in low-incidence areas.”)

to be more in line with their results  (Result “No significant differences were found in likelihood of being SARS-CoV-2 seropositive between participants belonging to the support services groups […] and those in the work-from-home group.”)  by removing the mention of potential increased seropositivity in support services from the abstract.